# Better Cone-Beam CT Artifact Correction via Spatial and Channel Reconstruction Convolution Based on Unsupervised Adversarial Diffusion Models

**DOI:** 10.3390/bioengineering12020132

**Published:** 2025-01-30

**Authors:** Guoya Dong, Yutong He, Xuan Liu, Jingjing Dai, Yaoqin Xie, Xiaokun Liang

**Affiliations:** 1Hebei Key Laboratory of Bioelectromagnetics and Neural Engineering, School of Health Sciences and Biomedical Engineering, Hebei University of Technology, Tianjin 300130, China; dongguoya@hebut.edu.cn (G.D.); yt.he1@siat.ac.cn (Y.H.); 2Shenzhen Institute of Advanced Technology, Chinese Academy of Sciences, Shenzhen 518055, China; xuan.liu@siat.ac.cn (X.L.); jj.dai@siat.ac.cn (J.D.); yq.xie@siat.ac.cn (Y.X.)

**Keywords:** CBCT reconstruction, diffusion model, deep learning

## Abstract

Cone-Beam Computed Tomography (CBCT) holds significant clinical value in image-guided radiotherapy (IGRT). However, CBCT images of low-density soft tissues are often plagued with artifacts and noise, which can lead to missed diagnoses and misdiagnoses. We propose a new unsupervised CBCT image artifact correction algorithm, named Spatial Convolution Diffusion (ScDiff), based on a conditional diffusion model, which combines the unsupervised learning ability of generative adaptive networks (GAN) with the stable training characteristics of diffusion models. This approach can efficiently and stably achieve CBCT image artifact correction, resulting in clear, realistic CBCT images with complete anatomical structures. The proposed model can effectively improve the image quality of CBCT. The obtained results can reduce artifacts while preserving the anatomical structure of CBCT images. We compared the proposed method with several GAN- and diffusion-based methods. Our method achieved the highest corrected image quality and the best evaluation metrics.

## 1. Introduction

Cone-Beam Computed Tomography (CBCT) is a medical imaging technology that uses cone-beam X-ray scanners and digital imaging techniques to obtain three-dimensional images. Compared to planning CT (pCT), CBCT offers greater real-time capabilities, shorter scan times, and lower X-ray doses. Therefore, CBCT holds significant clinical value in image-guided radiotherapy (IGRT). However, low-density soft tissues in CBCT images are often plagued with artifacts and noise, which can lead to missed diagnoses and misdiagnoses.

To reduce the impact of CBCT image artifacts on clinical diagnosis and improve diagnostic accuracy, researchers have proposed various methods. These methods can be divided into hardware-based correction methods and software-based correction. Due to the high cost and complex operation of hardware-based correction methods, software-based correction methods are preferred by researchers. In recent years, deep learning, which has developed rapidly in medical image processing [1,2], can be applied to image denoising [3], sparse reconstruction [4], artifact correction [5], and so on. Many deep learning-based methods have also been proposed in CBCT image artifact correction. Kida S et al. [6] applied U-Net to improve the spatial uniformity of CBCT images. Chang et al. [7] applied CNN to generate images with fewer artifacts directly from the sinogram domain, which can suppress the ring artifact effectively without the introduction of structure distortion.

However, the training of these deep learning methods often relies on a large number of paired data with or without artifacts, which is difficult to obtain in clinical scenarios. Recently, generative adversarial networks (GANs) [8] have made rapid progress in the field of image generation [9], style migration [10], and data augmentation [11]. Researchers try to introduce the unsupervised GAN-based methods into the CBCT image artifact correction task. Liang et al. [12] utilized a generative adversarial network framework with cycle-consistency (CycleGAN), which is capable of using unpaired CT and CBCT images to achieve image-to-image translation in an unsupervised manner. Dong et al. [13] used a multilayer and patch-based method to translate the low-quality CBCT images to high-quality CBCT images. Wang et al. [14] combined a double contrast learning adversarial network framework (DCLGAN) and post-processing techniques to obtain high-quality CBCT images. Image restoration methods based on GANs effectively reduce artifacts in CBCT images [12], and they can capture correlations in the data without requiring precisely labeled training data. This solves the problem of obtaining precise labels for medical images. However, the learning process of GANs may suffer from mode collapse, making them difficult to train [15]. In addition, images generated by GANs may have unreasonable or discontinuous parts in the overall structure, resulting in significant differences from the original images. Compared with other generative networks, GANs are over flexible, which makes simple GANs less controllable for larger images with more pixels.

To generate high-quality images and make the generation process more controllable, researchers have introduced diffusion model [16] into medical image restoration tasks. Ozbey M et al. [17] proposed an adversarial diffusion model named SynDiff to translate images between MRI and CT modalities. However, the network structure of SynDiff is complex, and this method will produce a gradient explosion in the training process. Li et al. [18] have proposed a Frequency-domain Guided Diffusion Model (FGDM) for image translation. However, this method only uses high-quality images and is based on an empirical frequency domain assumption without utilizing information from unpaired data.

This paper addresses the problems of artifact correction in CBCT images, with the following challenges: (i) It is difficult to obtain precisely labeled medical image training data; (ii) Most existing GAN- or diffusion-based methods require extensive computational resources, time, and large training datasets. To tackle the above challenges, we designed a novel CBCT image artifact correction method named the Spatial Convolution Diffusion (ScDiff) model. To solve the problem of obtaining accurate labels for medical images, the Match Module in ScDiff generates fake CBCT images based on cyclic-consistency loss, providing a reference for the Diffusive Module. The Diffusive Module is guided by the generated fake CBCT images and utilizes a large-step conditional diffusion process for efficient and accurate image sampling. In order to solve the problem of low training efficiency of diffusion models, we introduce the Spatial and Channel Reconstruction Convolution (SCConv) module [19] to the downsampling process of the Diffusive Module. This module can simultaneously process the spatial (shape, structure) and channel (depth) information of the image, making the network more refined and efficient in CBCT images generating.

We validated our network on the classical public TCIA lung datasets [20,21,22,23,24]. Compared to the most advanced method FGDM model, ScDiff showed significant improvements in metrics such as MAE, RMSE, PSNR, and SSIM.

In summary, the contributions of this paper are as follows:

(i) ScDiff combines the advantages of the unsupervised learning of GANs and the stable training characteristics of diffusion models. It enables stable and efficient training and synthesizing more realistic and clearer synthetic CBCT (sCBCT) images.

(ii) By introducing the SCConv module into ScDiff, the network training efficiency is effectively enhanced by reducing feature redundancy in network analysis.

(iii) We compared ScDiff with the SOTA method, Frequency-domain Guided Diffusion Model (FGDM). The results indicated that our method performs the best on all indicators.

## 2. Materials and Methods

### 2.1. Diffusion Model

The diffusion model transforms the original image into pure Gaussian noise by gradually adding noise in the forward process. In the reverse process, the diffusion model restores high-quality images from pure noise. By training the model to approximate the reverse process, we can generate complex data distributions from a simple noise distribution.

Given a clean image $x_{0}$, the forward diffusion process gradually adds Gaussian noise to the input image to form samples on the timesteps. By adding Gaussian noise gradually on the basis of the previous step $x_{t-1}$, the current time $x_{t}$ is obtained. This process can be regarded as a Markov chain. The transition probability of the state at the next moment depends only on its previous state. From $x_{t-1}$ to $x_{t}$, the mapping is as follows:(1)$$x_{t}=\sqrt{1-\theta_{t}}x_{t-1}+\sqrt{\theta_{t}}\epsilon ,\epsilon ∼N\left(0,I\right)$$
where *t* is the time step, t∈{0,1,2...T}. $\theta_{t}$ is the variance of the added noise for each step, which is an increasing sequence, ϵ is the added noise, which follows a unit Gaussian distribution, and $x_{t}$ is the data distribution of step *t*. The corresponding forward transition probability is as follows:(2)$$q\left(x_{t}|x_{t-1}\right)=N\left(x_{t};\sqrt{1-\theta_{t}}x_{t-1},\theta_{t}I\right)\left(t=1,2,\ldots ,T\right)$$
where $q\left(x_{t}|x_{t-1}\right)$ is the forward transition probability, N is the Gaussian distribution and *I* is the unit diagonal matrix. By repeatedly adding noise, any input image $x_{0}$ can be converted into a sample $x_{T}$ close to unit Gaussian noise.

In the reverse process, the task of the diffusion model is to remove the added noise and restore the original input image from a pure noise image. In the case of large *T* and small $\theta_{t}$, the posterior probabilities of $x_{t-1}$ and $x_{t}$ can be obtained from the Bayesian formula, which is an approximately Gaussian distribution:(3)$$p\left(x_{t-1}|x_{t}\right)=N\left(x_{t-1};\mu \left(x_{t},t\right),\Sigma \left(x_{t},t\right)\right)$$
where $p\left(x_{t-1}|x_{t}\right)$ is the reverse transition probability, $\mu (x_{t},t)$ is the mean and $\sum (x_{t},t)$ is the variance. They are both predicted by the model.

Usually, the variational lower bound (VLB) is used to train the diffusion model. The training goal is to minimize the KL divergence between the data distribution and the generation distribution. After model training, the generation process can gradually generate data from noise in the following ways:(4)$$x_{t-1}=\frac{1}{\sqrt{\gamma_{t}}}\left(x_{t}-\frac{\theta_{t}}{\sqrt{1-\bar{\gamma_{t}}}}\epsilon \left(x_{t},t\right)\right)\left(t=T,T-1,\ldots ,1\right)$$
where $\gamma_{t}=1-\theta_{t}$.

### 2.2. ScDiff

To improve the quality of CBCT, we propose an unsupervised conditional diffusion-based model. The network architecture is shown in Figure 1. The ScDiff model includes the Match Module and the Diffusive Module. The Match Module is used to generate fake CBCT images paired with CT images to realize unsupervised learning without the requirements of paired data.

The Diffusive Module improves the forward process of a traditional diffusion model. The traditional diffusion model usually needs a large time *t* to ensure that the step is small enough to meet the normality assumption [25]. This method will limit the efficiency of image generation. Specifically, we use a large step *k* in the Diffusive Module to achieve an efficient sampling process. In the reverse process of the Diffusive Module, the generator first denoising the current noisy image to obtain a prediction as close to the clean image as possible.

***Match Module*** Given unpaired CBCT and pCT images $x^{cbct}$ and $x^{ct}$. We obtain $x_{0}^{cbct}$ and $x_{0}^{ct}$ by random sampling. Then, we use a generator $G_{\beta }$ to translate pCT images to fake CBCT images and other $G_{\beta }^{\prime }$ to translate fake CBCT images back to pCT images. The generated fake CBCT images are denoted as ${\hat{y}}^{cbct}$:(5)$$\begin{array}{cc} {\hat{y}}_{0}^{cbct}=G_{\beta }\left(x_{0}^{ct}\right),\; & {\hat{y}}_{0}^{ct}=G_{\beta }^{\prime }\left({\hat{y}}_{0}^{cbct}\right), \end{array}$$

Two discriminators $D_{\beta }$ and $D_{\beta }^{\prime }$ are used for judging the authenticity of the generated ${\hat{y}}_{0}^{cbct}$ and ${\hat{y}}_{0}^{ct}$. For $G_{\beta },D_{\beta }$ and $G_{\beta }^{\prime },D_{\beta }^{\prime }$, unsaturated adversarial loss [26] is adopted:(6)$$\begin{array}{cc} \; & L_{G_{\beta }}=E_{p_{\beta }\left(x_{0}^{cbct}|x_{0}^{ct}\right)}\left[-\log\left(D_{\beta }\left({\hat{y}}_{0}^{cbct}\right)\right)\right]\\ \; & L_{G_{\beta }^{\prime }}=E_{p_{\beta }^{\prime }\left(x_{0}^{ct}|{\hat{y}}_{0}^{cbct}\right)}\left[-\log\left(D_{\beta }^{\prime }\left({\hat{y}}_{0}^{ct}\right)\right)\right] \end{array}$$(7)$$\begin{array}{cc} \; & L_{D_{\beta }}=E_{q\left(x_{0}^{cbct}|x_{0}^{ct}\right)}\left[-\log\left(D_{\beta }\left(x_{0}^{cbct}\right)\right)\right]+E_{p_{\beta }\left(x_{0}^{cbct}|x_{0}^{ct}\right)}\left[-\log\left(1-D_{\beta }\left({\hat{y}}_{0}^{cbct}\right)\right)\right]\\ \; & L_{D_{\beta }^{\prime }}=E_{q\left(x_{0}^{ct}|{\hat{y}}_{0}^{cbct}\right)}\left[-\log\left(D_{\beta }^{\prime }\left(x_{0}^{ct}\right)\right)\right]+E_{p_{\beta }^{\prime }\left({\hat{y}}_{0}^{ct}|x_{0}^{cbct}\right)}\left[-\log\left(1-D_{\beta }^{\prime }\left({\hat{y}}_{0}^{ct}\right)\right)\right] \end{array}$$
where $p_{\beta }\left(x_{0}^{cbct}|x_{0}^{ct}\right)$ and $p_{\beta }^{\prime }\left({\hat{y}}^{ct}|x_{0}^{cbct}\right)$ mean the network parametrization of the conditional distribution of $x_{0}^{cbct}$ and ${\hat{y}}^{ct}$ given the $x_{0}^{ct}$ and $x_{0}^{cbct}$ image. $q\left(x_{0}^{cbct}|x_{0}^{ct}\right)$ and $q\left(x_{0}^{ct}|{\hat{y}}_{0}^{cbct}\right)$ represent the true conditional distribution of the image obtained by the generator.

In order to ensure the consistency between the fake CBCT images and pCT images, we use the cycle-consistency loss to constrain the performance of the model. Comparing the images ${\hat{y}}_{0}^{ct}$ generated by the Match Module with the input $x_{0}^{ct}$, the cycle-consistency loss function is obtained:(8)$$\begin{array}{c} \begin{array}{c} L_{\mathrm{cycM}}=E_{t,q\left(x_{0}^{ct}\right)}(\lambda_{1}|x_{0}^{ct}-{\hat{y}}_{0}^{ct}{|}_{1}) \end{array} \end{array}$$
where ${\hat{y}}_{0}^{ct}$ is obtained from $G_{\beta }^{\prime }\left({\hat{y}}_{0}^{cbct}\right)$. $\lambda_{1}$ is the weight of the cycle-consistency loss item of the Match Module. The $\ell_{1}$-norm of the difference between two images is used as a consistency measure [27].

The loss of the Match Module is as follows: $L_{Match}=L_{G_{\beta }}+L_{G_{\beta }^{\prime }}+L_{D_{\beta }}+L_{D_{\beta }^{\prime }}$

***Diffusive Module*** In this module, we used the fake CBCT images generated by the Match Module as a condition to predict its corresponding sCBCT images. In the forward process of the Diffusive Module, we use *k* as a time step and add Gaussian noise to $x_{0}^{ct}$ step by step until T. We can obtain different $\left\{x_{0}^{ct},x_{k}^{ct}...x_{t-k}^{ct},x_{t}^{ct}...x_{T}^{ct}\right\}$ with different levels of noise. Then we input $\left(x_{t}^{ct},{\hat{y}}_{0}^{cbct}\right)$ and current time t∼U({0,k,…,T}) to generator $G_{\alpha }\left(x_{t}^{ct},{\hat{y}}_{0}^{cbct},t\right)$. Each *k* step generates a deterministic estimate ${\hat{y}}_{t-k}^{scbct}$ of the pCT image. After the iterations, we obtain ${\hat{y}}_{0}^{scbct}$. The transition probability is $q\left(x_{t-k}^{ct}|x_{t}^{ct},{\hat{y}}_{0}^{cbct}\right)$. After the discrimination of the discriminator $D_{\alpha }$, the final result closest to the real noiseless target image is obtained. $G_{\alpha }$ uses unsaturated counter loss [26], $D_{\alpha }$ uses unsaturated counter loss with gradient penalty [28]:(9)$$L_{G_{\alpha }}=E_{t,q\left(x_{t}^{ct}|x_{0}^{ct},{\hat{y}}_{0}^{cbct}\right),p_{\alpha }\left(x_{t-k}^{ct}|x_{t}^{ct},{\hat{y}}_{0}^{cbct}\right)}\left[-\log\left(D_{\alpha }\left({\hat{y}}_{t-k}^{scbct}\right)\right)\right]$$(10)$$\begin{array}{c} L_{D_{\alpha }}=E_{t,q\left(x_{t}^{ct}|x_{0}^{ct},{\hat{y}}_{0}^{cbct}\right)}\left[E_{q\left(x_{t-k}^{ct}|x_{t}^{ct},{\hat{y}}_{0}^{cbct}\right)}\left[-\log\left(D_{\alpha }\left(x_{t-k}^{ct}\right)\right)\right]\right.\\ +E_{p_{\alpha }\left(x_{t-k}^{ct}|x_{t}^{ct},{\hat{y}}_{0}^{cbct}\right)}\left[-\log\left(1-D_{\alpha }\left({\hat{x}}_{t-k}^{scbct}\right)\right)\right]\\ \left.+\eta E_{q\left(x_{t-k}^{ct}|x_{t}^{ct},{\hat{y}}_{0}^{cbct}\right)}{∥\nabla_{x_{t-k}^{ct}}D_{\alpha }\left(x_{t-k}^{ct}\right)∥}_{2}^{2}\right] \end{array}$$
where η is the weight of the gradient penalty.

In order to ensure the consistency between the sCBCT images generated by the Diffusive Module and pCT images, we use the cycle-consistency loss to constrain the performance of the Diffusive Module. Comparing the images ${\hat{y}}_{0}^{scbct}$ with the $x_{0}^{ct}$ in the end of every step, the cycle-consistency loss function is obtained:(11)$$\begin{array}{c} \begin{array}{c} L_{\mathrm{cycD}}=E_{t,q(x_{t}^{ct}|x_{0}^{ct})}(\lambda_{2}|x_{0}^{ct}-{\hat{y}}_{0}^{scbct}{|}_{1}) \end{array} \end{array}$$
where ${\hat{y}}_{0}^{scbct}$ is obtained from $G_{\alpha }\left(x_{t}^{ct},{\hat{y}}_{0}^{cbct},t\right)$. $\lambda_{2}$ is the weight of the cycle-consistency loss item of the Match Module. The $\ell_{1}$-norm of the difference between two images is used as a consistency measure [27].

The loss of Diffusive Module is $L_{Diff}=L_{G_{\alpha }}+L_{D_{\alpha }}$, and the cycle-consistency loss of the ScDiff is $L_{cyc}=L_{\mathrm{cycM}}+L_{\mathrm{cycD}}$

The Match Module and Diffusive Module are trained jointly. The total loss function of the ScDiff is as follows:(12)$$L_{\mathrm{total}\;}=\lambda_{3}L_{Match}+\lambda_{4}L_{Diff}+L_{\mathrm{cyc}}$$
where $\lambda_{3},\lambda_{4}$ are the weight of the adversarial loss term of the Match Module and the Diffusion Module. The Match Module provides paired predictive images for the Diffusion Module during the training process. During inference process, Diffusion Module only needs to execute the generator $G_{\alpha }$. Starting from time *T*, the generator $G_{\alpha }$ gradually obtains the target image step by step and uses the result of the previous step as an input sample for the next step. Finally, output clean sCBCT images ${\hat{y}}^{scbct}$.

***SCConv Module*** In order to reduce feature redundancy and save computational costs, we introduce SCConv [19] in the downsampling process of generator $G_{\alpha }$ as shown on the right side of Figure 1. This module consists of two parts, Spatial Reconstruction Unit (SRU) and Channel Reconstruction Unit (CRU). When inputting $\left(x_{t}^{ct},{\hat{y}}_{0}^{cbct}\right)$, the feature $f_{0}$ is obtained through a 3×3 convolution and a ResnetBlock module. SRU first performs group normalization (GN) on feature $f_{0}$:(13)$$f=GN\left(f_{0}\right)=\gamma \frac{f_{0}-\mu }{\sqrt{\sigma^{2}+\varepsilon }}+z$$
where μ and σ are the mean and standard deviation of $f_{0}$. ε is a small positive number added for stable division, while γ and *z* are trainable affine transformations. γ is used to measure the spatial pixel variance of each batch and channel. The larger the γ, the more spatial pixels change, and the richer the spatial information. We can obtain the normalized correlation weight $W_{\gamma }$ representing the importance of different feature maps through Equation (Equation 14). Mapping $W_{\gamma }$ to the range (0,1) using the sigmoid function and gate it with a threshold of 0.5.(14)$$W_{\gamma }=\left\{w_{i}\right\}=\frac{\gamma_{i}}{\sum_{j=1}^{C}\gamma_{j}},\;i,j=1,2,\cdots ,C$$
where *C* is the channel of *f*. The weight greater than the threshold is set to 1 to obtain the information weight $W_{1}$, and the weight less than the threshold is set to 0 to obtain the redundant weight $W_{2}$. Multiply the information weight and redundant weight with *f* element by element to obtain $f_{1}^{w}$, which has informative and expressive spatial contents, and $f_{2}^{w}$, which has little or no information:(15)$$\begin{array}{cc} \; & f_{1}^{w}=W_{1}⊗f\\ \; & f_{2}^{w}=W_{2}⊗f \end{array}$$
where ⊗ represents element-wise multiplication. In order to fully combine the weighted two different information rich features and enhance the information flow between them, cross reconstruction [29] is used to obtain $f^{w_{1}}$ and $f^{w_{2}}$. Finally, connect $f^{w_{1}}$ and $f^{w_{2}}$ to obtain $f^{w}$.(16)$$f^{w}=f^{w_{1}}\cup f^{w_{2}}$$

The representative features in *f* are enhanced in $f^{w}$, and the redundancy of *f* in the spatial dimension is suppressed. However, there is still channel redundancy in $f^{w}$. So, CRU first performs a split operation to handle the channel redundancy in $f^{w}$. Divide $f^{w}$ into vC with *v* channel and (1−v)C with 1−v channels. Channel compression is performed through 1×1 convolution to obtain $f_{vC}^{w}$ and $f_{(1-v)C}^{w}$.

Next, use a 3×3 groupwise convolution (GWC) [30] with g = 2 and a 1×1 pointwise convolution (PWC) [31] to extract information from $f_{vC}^{w}$. Add the output results together to form $Y_{1}^{C}$. Extract shallow details from $f_{(1-v)C}^{w}$ using 1×1 PWC as a supplement to $f_{vC}^{w}$. Connect the output of PWC with $f_{(1-v)C}^{w}$ to obtain $Y_{2}^{C}$. Finally, we adaptively merge $Y_{1}^{c}$ and $Y_{2}^{c}$ using a simplified SKNet method [32] to obtain the final output *Y*.

The SCConv module reduces the spatial and channel redundancy of input features through SRU and CRU, effectively reducing computational costs and improving model performance.

## 3. Experiments

### 3.1. Experimental Settings

***Datasets.*** The data collection is made up of images from 20 patients with locally advanced non-small cell lung cancer taken throughout chemotherapy and radiation therapy. These images are sourced from TCIA [20,21,22,23,24], with undergoing CBCT and pCT scans for each patient. After registering and filtering out poor-quality data, a total of 6721 slices of CBCT and pCT images were obtained. For training, 16 patients contributed 5377 slices. Four patients provided 1344 slices for testing. The axial matrix size of CBCT and pCT images is 512×512. All images in the training dataset were normalized to the range [−1, 1].

***Training and Inference.*** This model is trained by using an NVIDIA RTX 3090 with 24 GB of RAM. Training was carried out using the Adam optimization technique with β1 set to 0.5 and β2 set to 0.9, as these parameter values have been demonstrated to work well in similar deep learning tasks and can help the model converge stably during the training process. On the test set in each dataset, the model’s performance was assessed. After finishing the network training, the sCBCT images were produced. The model took 360 h for training. Meanwhile, during training, it took about 30 min for the diffusion model to conduct image production training for each test patient round.

***Evaluation Metric.*** For quantitative assessment, lung cancer patient image pairs from CBCT and pCT were utilized. The reference was the pCT. A comparison of the axial views of pCT and CBCT images was made at the same window level in order to confirm that our strategy has improved the quality of synthetic pCT images. The structural similarity and spatial uniformity of the generated images, as well as the improvement of sCBCT over CBCT, were statistically evaluated using the mean absolute error (MAE) [33], root mean square error (RMSE) [33], peak signal-to-noise ratio (PSNR) [34] and structural similarity index (SSIM) [35]. The following are the definitions of these metrics between sCBCT and pCT:(17)$$\begin{array}{cc} MAE & =\frac{1}{M}\sum_{i,j}^{n_{sCBCT}n_{pCT}}\left|sCBCT\left(i,j\right)-pCT\left(i,j\right)\right|,\\ RMSE & =\sqrt{\frac{1}{n_{sCBCT}n_{pCT}}\sum_{i,j}^{n_{sCBCT}n_{pCT}}{\left(sCBCT\left(i,j\right)-pCT\left(i,j\right)\right)}^{2}},\\ PSNR & =10\log_{10}\left(\frac{MAX^{2}}{\sum_{i,j}^{n_{sCBCT}n_{pCT}}\frac{{\left(sCBCT\left(i,j\right)-pCT\left(i,j\right)\right)}^{2}}{n_{sCBCT}n_{pCT}}}\right),\\ SSIM & =\frac{\left(2\mu_{sCBCT}\mu_{pCT}+c_{1}\right)\left(2\sigma_{sCBCT\cdot pCT}+c_{2}\right)}{\left(\mu_{sCBCT}^{2}+\mu_{pCT}^{2}+c_{1}\right)\left(\sigma_{sCBCT}^{2}+\sigma_{pCT}^{2}+c_{2}\right)}. \end{array}$$

***Competing Methods.*** We compared several SOTA GAN - and diffusion-based methods with ScDiff. All competing methods use unpaired CBCT and pCT for unsupervised learning. We adjusted the hyperparameters of each method to improve the performance of the validation set. The adjusted parameters include epoch, learning rate, and loss weight. The hyperparameters of ScDiff were 50 epochs, 2 × $10^{-5}$ learning rate, T = 1000, a step size of k = 250, and diffusion steps T/k = 4. Weights for cycle-consistency were $\lambda_{3}$, $\lambda_{4}$ = 0.5. The hyperparameters of CycleGAN, CUT, and DCLGAN were 100 epochs, a $10^{-4}$ learning rate linearly decayed to 0 in the last 50 epochs. The hyperparameters of SynDiff were 50 epochs, $10^{-4}$ learning rate, T = 1000, a step size of k = 250, and diffusion steps T/k = 4. Weights for losses were $\lambda_{1\phi ,1\theta }$ = 0.5 and $\lambda_{2\phi ,2\theta }$ = 1. The hyperparameters of FGDM were 50 epochs, $10^{-4}$ learning rate, T = 1000, k = 1, and 1000 diffusion steps. Weight for loss was 1.

### 3.2. Results

Figure 2 illustrates the axial, sagittal, and coronal views of two patients’ CBCT, sCBCT, and pCT images. It can be observed that the stripe artifacts in the CBCT image are severe, leading to partial tissue loss and significant CT value variations. In comparison to CBCT, the generated sCBCT image demonstrates effective artifact suppression and noticeable improvements in soft tissue contrast, spatial uniformity, and clarity.

Furthermore, Figure 3 presents the axial view of a selected patient, with the regions of interest (ROI) region outlined in red and green. It is evident that the quality of the sCBCT is significantly superior to the CBCT image, with image quality and spatial uniformity being enhanced while anatomical consistency is maintained. The third row focuses on skeletal structures, and despite registration issues resulting in slight anatomical discrepancies between CBCT and pCT, the anatomical consistency between sCBCT and CBCT images indicates faithful restoration of CBCT’s anatomical information. The violin plot shows that the sCBCT image is closer to the pCT image in terms of HU values, indicating that the proposed method produces more realistic images.

The distribution of HU values is illustrated in Figure 4, with the red line representing the longitudinal distribution and the yellow line representing the transverse distribution. The results show that, compared to CBCT, sCBCT exhibits HU values that are closer to those of pCT, indicating that sCBCT can not only reduce artifacts but also correct HU value discrepancies.

Figure 5 and Table 2 illustrate the qualitative comparison and quantitative comparison of the results obtained by different methods, respectively. In terms of visual effects, all models can achieve the elimination of artifacts compared to CBCT. Specifically, the images generated by CycleGAN and CUT seem to be satisfactory. However, as can be seen from the zoomed-in green box, these two methods cannot completely eliminate stripe artifacts and will result in structural losses, such as soft tissue and spinal regions. Additionally, there is a noticeable blurring of the boundaries of soft tissue and a significant CT value error. FGDM exhibits a less effective image restoration under the same number of iterations and hyperparameters, particularly for high-contrast anatomical features like the skeleton. Relatively, images generated by DCLGAN and SynDiff better preserve anatomical details, striking a balance between realism and fidelity, although they still fall short compared to ScDiff. As highlighted in the green box, ScDiff eliminates stripe artifacts and effectively retains structural details compared to other methods. Even though ScDiff works the best out of all the techniques, there are still a few small structural inconsistencies, with the source images needing to be worked out and optimized. From the difference plot on the right, the difference between ScDiff and pCT is the smallest. This indicates that the proposed method can effectively reduce the difference between CBCT and pCT. The obtained results can preserve more anatomical structures and have a better effect on removing artifacts.

The quantitative evaluation results are presented in Table 1, along with the average values. This covers the CBCT, sCBCT, and pCT compared to pCT MAE, RMSE, PSNR, and SSIM values. Spatial consistency and structural similarity around pCT are indicated by sCBCT’s MAE value of 19.398 HU and RMSE value of 62.707 HU, respectively, compared with CBCT. After correction, PSNR increases from 24.540 dB to 30.469 dB, and SSIM increases from 0.811 to 0.924, suggesting that the generated sCBCT images are of lower distortion and better similarity to pCT in terms of brightness, contrast, and structure, demonstrating the effectiveness of our denoising approach.

As demonstrated in Table 2, in a quantitative comparison of CycleGAN, CUT, DCLGAN, SynDiff, and FGDM, CycleGAN and DCLGAN perform less well in RMSE but comparatively better in SSIM. In general, additional loss functions used by CycleGAN, CUT, and DCLGAN to preserve structure have a negative impact on image quality and lower PSNR. Although SynDiff has shown overall good results in CBCT artifact correction tasks, it still lags behind ScDiff in PSNR and SSIM. FGDM, another diffusion model-based approach, has a low SSIM of only 0.86, indicating poor performance. All things considered, ScDiff performs better than other techniques on every metric. This is attributed to its capacity to inherit both the realistic and high-quality image generation ability of the diffusion model and the image preservation capabilities of GAN.

After the correction of ScDiff, the quality of CBCT images has significantly improved, with clearer anatomical structures and higher contrast. Doctors can observe and evaluate lesion areas more accurately through corrected sCBCT images and develop more precise treatment plans. The corrected sCBCT image plays a crucial role in image-guided radiotherapy (IGRT). It can significantly improve the accuracy of positioning and target area delineation, laying the foundation for precision radiotherapy.

### 3.3. Ablation Study

In ScDiff, cycle-consistency loss plays a role in improving generated image quality and ensuring generated image consistency. Meanwhile, ScDiff adopts the SCConv module to address the limitations of traditional convolution operations in both spatial and channel dimensions [19]. To test their contributions, we conducted a study of ScDiff through the ablation of different modules in this sub-section. The results of the ablation study are shown in Table 3. It can be seen that the introduction of cycle-consistency loss maintains the ability of high-quality image generation, while the SCConv module improves the efficiency of network training and testing.

## 4. Discussion

We proposed an unsupervised CBCT artifact correction method based on conditional diffusion, ScDiff, which includes the Match Module and Diffusive Module. The Match Module introduces cycle-consistency loss in two generator-projector pairs to generate CBCT images paired with the target sCBCT images. The Diffusive Module receives the output of the Match Module as a guide and uses the CT image as the input to generate the sCBCT image. The forward process adds noise to the CT image. In the inverse process of diffusion, the generator first denoises the current noisy image to obtain a prediction as close to the clean image as possible.

The task of this paper is essentially a medical image translation task, and the size and specificity of the dataset used for training are limited. The output of the model depends on the CBCT image in anatomical structure. This requires the Match Module to output excellent CBCT images. This conclusion is verified in the experiment. If the output obtained from the Match Module shows underfitting during training and is used for subsequent steps, the final result cannot achieve the best effect. In addition, we found that gradient explosion occurred in the Diffusive Module during the training process. This is due to the high learning rate. And the scale of noise increases or decreases gradually in the process of forward and reverse denoising of the diffusion model. If the scale of noise is not adjusted properly in this process, it may lead to too large a gradient at high time steps, which makes it difficult to train the model stably. In order to solve this problem, we compared the training loss of the ScDiff model at different learning rates and the training loss and validation loss of ScDiff under different dropout rates. As shown in Figure 6. When lr is set to 1 × $10^{-3}$, a gradient explosion phenomenon occurs during the model training process. When lr is set to 1 × $10^{-4}$, the model requires more epochs for convergence. So, we ultimately used lr = 5 × $10^{-4}$ as the parameters for model training. However, overfitting occurred during the validation process, as shown in the first column of Figure 6b when Dropout = 0. To address the issue of overfitting, we set the dropout values to 0.2 and 0.5, respectively. The results show that the model can fit normally when dropout = 0.2.

Although the proposed ScDiff has shown encouraging performance in correcting scattering artifacts in CBCT, there are still some problems to be solved. When removing fringe artifacts, a small number of small lung textures in the feature image are ignored. This may lead to missed diagnosis or misdiagnosis. In addition, a CBCT scanner with a shorter scanning time may be used in clinical. The singing of the breathing cycle and the shortening of the projection interval will lead to a further decline in image quality. The current experiment only considers a single scanning scenario. The robustness of this method needs to be verified. In the future, we will obtain clinical data from different parts and scan parameters or simulate projections from different views for further experiments.

## 5. Conclusions

This study proposes a diffusion model for the artifact correction task of lung CBCT images. ScDiff achieves unsupervised learning by integrating the Match Module and the Diffusive Module into a cycle-consistency architecture while utilizing a large-step conditional diffusion process for efficient and accurate image sampling. In addition, the model introduces a SCConv module to handle redundant features, thereby improving network performance. Compared with existing artifact correction methods, the ScDiff model exhibits excellent performance and can generate high-fidelity and high-contrast images.

## Figures and Tables

**Figure 1 bioengineering-12-00132-f001:**
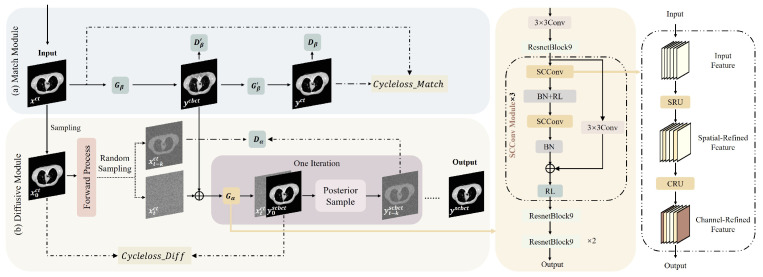
An overview of the ScDiff framework: (**a**) Match Module. (**b**) Diffusive Module. Each purple block shows one iteration for calculating ${\hat{y}}_{t-k}^{scbct}$ from $x_{t}^{ct}$ while $x_{t}^{ct}$ is sampled from a unit Gaussian distribution. The right part of the figure shows the details of our proposed SCConv Module.

**Figure 2 bioengineering-12-00132-f002:**
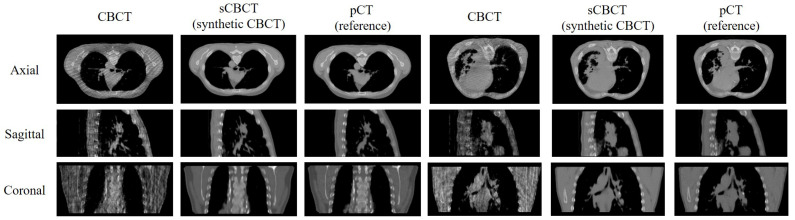
Comparisons of the image quality between CBCT, sCBCT (proposed method), and pCT (reference) during a particular patient phase. The images are sagittal, coronal, and axial in the top, middle, and bottom rows, respectively. The CBCT, sCBCT (proposed method), and pCT (reference) are indicated by the left, center, and right, respectively.

**Figure 3 bioengineering-12-00132-f003:**
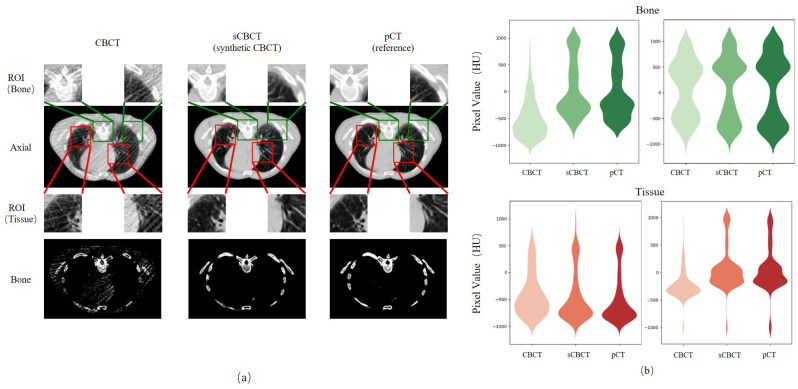
Comparison of HU values in ROI. (**a**) Red-colored boxes in the images are zoomed to demonstrate the tissue and green-colored boxes in the images are zoomed to demonstrate the bone. The range of the CT number display window is [−1000, 1000] HU. The fourth row simply shows the bone structure, excluding the soft tissue. The window for display is [500, 750] HU. (**b**) Comparison of HU values in ROI with Violin diagram.

**Figure 4 bioengineering-12-00132-f004:**
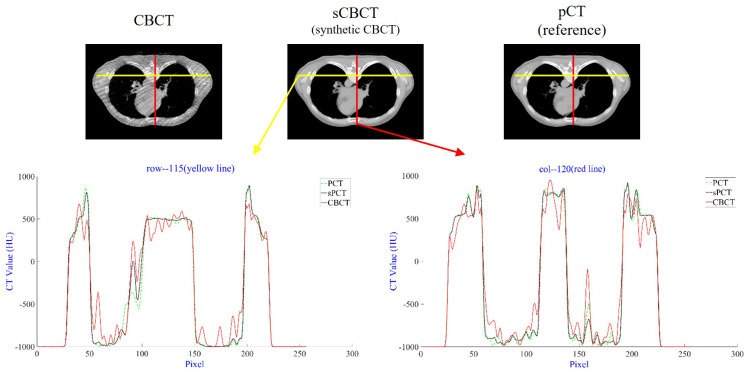
In the first row, CBCT, sCBCT, and pCT are displayed in axial perspectives from left to right. The HU value distributions for CBCT, sCBCT, and pCT are shown in the second row along yellow lines (line 115) and red lines (line 120). The CBCT, sCBCT, and pCT HU value distributions are shown by the red, black, and green lines, respectively. The display window’s current setting is [−1000, 1000] HU.

**Figure 5 bioengineering-12-00132-f005:**
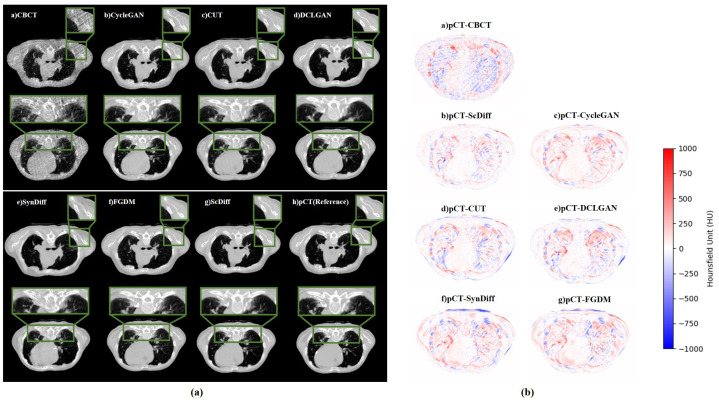
(**a**) Comparison of the image quality produced by using various methods. The inference sample is the pCT. The green box displays local zoomed-in images of the results obtained by different methods. The display window is [−1000, 1000] HU. (**b**) Difference images between pCT and different methods.

**Figure 6 bioengineering-12-00132-f006:**
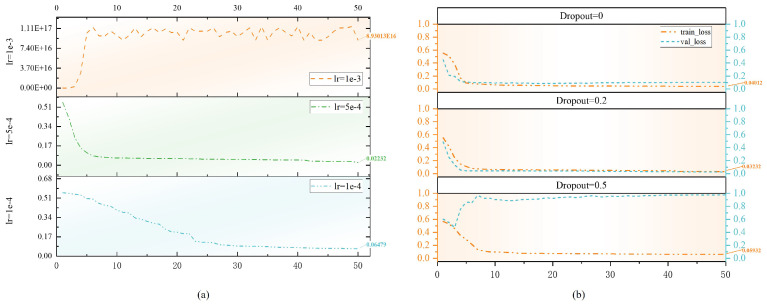
(**a**) Training loss curves of ScDiff model at different learning rates. (**b**) Training and validation loss curves of ScDiff model at different dropout rates.

**Table 1 bioengineering-12-00132-t001:** Comparison results for each patient in test datasets with MAE (HU), RMSE (HU), PSNR (dB), and SSIM.

Data	Image Types	MAE↓	RMSE↓	PSNR↑	SSIM↑
Patient 1	sCBCT-pCT	18.496	60.093	30.662	0.926
Patient 2	sCBCT-pCT	18.319	59.662	30.753	0.927
Patient 3	sCBCT-pCT	20.803	67.172	30.263	0.921
Patient 4	sCBCT-pCT	19.852	63.899	30.196	0.923
Mean	sCBCT-pCT	**19.398**	**62.707**	**30.469**	**0.924**
	CBCT-pCT	49.945	117.803	24.540	0.811

Remark: “sCBCT” denotes synthetic CBCT after correction. ↑ (↓) indicates that the larger (smaller) the value, the better the performance; the **Bold** numbers indicate this metric’s best performance.

**Table 2 bioengineering-12-00132-t002:** Quantitative comparison of the image quality produced using various techniques.

	Methods	MAE↓	RMSE↓	PSNR↑	SSIM↑	Inference Time↓
	CycleGAN [12]	36.497	97.483	26.510	0.853	**0.236**
GAN-Based	CUT [13]	37.764	101.219	25.991	0.850	0.258
	DCLGAN [14]	32.491	88.273	27.227	0.876	0.278
	SynDiff [17]	34.845	94.521	26.730	0.862	0.312
Diffusion-Based	FGDM [36]	31.170	88.825	27.323	0.881	0.397
	**ScDiff**	**19.398**	**62.707**	**30.469**	**0.924**	**0.228**

Remark: ↑ (↓) indicates that the larger (smaller) the value, the better the performance; the **Bold** and underlined numbers indicate this metric’s best and second-best performance, respectively.

**Table 3 bioengineering-12-00132-t003:** Average inference time per slice and quantitative comparison of the image quality.

Baseline	Cycle-Consistency Loss	SCConv	Perceptual Loss	Inference Time (s) ↓	PSNR↑	SSIM↑
*√*				0.312	24.412	0.736
*√*	*√*			0.342	29.330	0.862
*√*		*√*		**0.194**	24.711	0.753
*√*	*√*	*√*		0.228	**29.976**	**0.919**
*√*	*√*	*√*	*√*	0.237	20.853	0.615

Remark: ↑ (↓) indicates that the larger (smaller) the value, the better the performance; the **Bold** and underlined numbers indicate this metric’s best and second-best performance, respectively.

## Data Availability

https://www.cancerimagingarchive.net/.

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
