# Peer review of "Better Cone-Beam CT Artifact Correction via Spatial and Channel Reconstruction Convolution Based on Unsupervised Adversarial Diffusion Models"

_bioengineering, 2025, doi:10.3390/bioengineering12020132_

Round 1
Reviewer 1 Report
Comments and Suggestions for Authors
How does the proposed unsupervised adversarial diffusion model improve upon traditional artifact correction techniques?
What specific spatial and channel reconstruction methods are utilized in the convolution process of this study?
What metrics were used to evaluate the effectiveness of the artifact correction in the experiments conducted?
How does the use of an unsupervised approach enhance the model's adaptability to various imaging scenarios compared to supervised methods?
Comments on the Quality of English LanguageHow does the proposed unsupervised adversarial diffusion model improve upon traditional artifact correction techniques?
What specific spatial and channel reconstruction methods are utilized in the convolution process of this study?
What metrics were used to evaluate the effectiveness of the artifact correction in the experiments conducted?
How does the use of an unsupervised approach enhance the model's adaptability to various imaging scenarios compared to supervised methods?
Author Response
Comments 1: How does the proposed unsupervised adversarial diffusion model improve upon traditional artifact correction techniques?
Response 1: Thanks for raising this important issue. Firstly, the unsupervised adversarial diffusion model we propose is an end-to-end learning model that can automatically learn the distribution of artifacts and correction strategies. This method has stronger adaptability and better generalization ability. Secondly, the unsupervised characteristic of the model doesn’t rely on paired training data, which reducing the cost of data tagging. Finally, the progressive denoising process of the model makes the generated results more stable.
Comments 2: What specific spatial and channel reconstruction methods are utilized in the convolution process of this study?
Response 2: Thanks for the comment. We have provided a detailed introduction in the SCConv module of section 2.2 and figure 1.
Comments 3: What metrics were used to evaluate the effectiveness of the artifact correction in the experiments conducted?
Response 3: Thanks for the comment. We evaluate the quality of images after artifact correction by mean absolute error(MAE), root mean square error(RMSE), peak signal-to-noise ratio(PSNR) and structural similarity index(SSIM). The relevant formulas can be found in the Experimental Settings of Section 3.1.
Comments 4: How does the use of an unsupervised approach enhance the model's adaptability to various imaging scenarios compared to supervised methods?
Response 4: Thanks for pointing out this problem. Compared with supervised methods, the unsupervised model proposed in this paper uses a contrastive learning framework. By comparing learning methods, the model can learn the intrinsic representation and semantic information of images. The contrastive learning of positive and negative sample pairs from different perspectives enables the model to capture the essential features of images without being affected by changes in image conditions, significantly improving the model's scene generalization ability.

Reviewer 2 Report
Comments and Suggestions for Authors
Author have presented a method for Cone-Beam CT Artifact Correction using Convolution Based Unsupervised Adversarial Diffusion Models with Spatial and Channel Reconstruction.
Paper is well written.
Here are few suggestions
Many grammatical mistakes in manuscripts.
Please confirm is it Modules or Models in section 2.2 for Match and Diffusive (In my opinion it should be Models)
What is the hyperparameter setting
Please also use perceptual loss for comparison
Provide the mathematical framework for quantitative metrics used.
Comments on the Quality of English Language
Many grammatical mistakes in manuscripts.
Author Response
Comments 1: Many grammatical mistakes in manuscripts.
Response 1: Thanks for your suggestion. We have tried our best to polish the language in the revised manuscript.
Comments 2: Please confirm is it Modules or Models in section 2.2 for Match and Diffusive (In my opinion it should be Models).
Response 2: Thanks for pointing out this problem. I think here should be module. The Match Module and Diffusive Module form the ScDiff Model together.
Comments 3: What is the hyperparameter setting.
Response 3: Thanks for the comment. Hyperparameters are configuration variables that control the training process of a model, such as learning rate, step size epochs. We set the hyperparameters to achieve the best performance of the model. We have added the detailed hyperparameter of each method in section 3.1, Competing Methods.
Comments 4: Please also use perceptual loss for comparison.
Response 4: We feel great thanks for your professional review work on our article. We have added the perceptual loss to the total loss function. The results is shown in Table 3.
Comments 5: Provide the mathematical framework for quantitative metrics used.
Response 5: Based on your comments, we have added the definitions of evaluation metrics in Experimental Settings of section 3.1.

Reviewer 3 Report
Comments and Suggestions for Authors
This paper proposes a new method to correct blurred or distorted parts (artifacts) in medical imaging. Cone-Beam CT (CBCT) is used in radiation therapy, but artifacts in soft tissues can cause diagnostic errors. The research team developed a model called ScDiff, which combines GAN (image generation technology) and Diffusion Model (a technique for stable image restoration). This model enhances CBCT images without requiring paired data and preserves anatomical structures. It generates clearer and more accurate images than existing methods, aiding in better diagnosis during radiation therapy.
1. ScDiff, which combines the Diffusion Model and GAN, requires high computational costs. The paper mentions that training took 360 hours using an NVIDIA 3090 GPU. Research on model optimization and compression is necessary. Additionally, comparative experiments on training and inference speeds should be conducted to demonstrate how ScDiff's efficiency compares to other models.
2. While the paper evaluates ScDiff's performance using quantitative metrics such as PSNR and SSIM, there is a lack of detailed discussion on its clinical significance. The authors should assess how doctors have applied ScDiff-generated images in real diagnoses or how it affects diagnostic accuracy. Comparing results with actual diagnoses or incorporating feedback from medical professionals would strengthen the model’s practical effectiveness.
3. The paper mentions a gradient explosion issue during the training of the Diffusion Model. Although the authors resolved this by adjusting the learning rate (lr = 5e-4) and applying dropout (0.2), there is no clear explanation for why these specific values were chosen. A more detailed justification of the parameter selection process is needed.
4. The comparison between ScDiff and existing models (e.g., CycleGAN, DCLGAN, SynDiff) relies mainly on quantitative metrics such as PSNR and SSIM. To enhance visual evidence, the paper should highlight how specific artifacts (e.g., ring artifacts, linear artifacts) were corrected using zoomed-in images with annotated arrows.
5. The SCConv (Spatial and Channel Reconstruction Convolution) module is a critical part of the model, but its implementation details are described rather briefly. The paper should provide a block diagram illustrating the SCConv module’s structure and operational principles in greater detail.
Author Response
Comments 1: ScDiff, which combines the Diffusion Model and GAN, requires high computational costs. The paper mentions that training took 360 hours using an NVIDIA 3090 GPU. Research on model optimization and compression is necessary. Additionally, comparative experiments on training and inference speeds should be conducted to demonstrate how ScDiff's efficiency compares to other models.
Response 1: Based on your suggestion, we have added inference time as an evaluation metric to table 2. The training time of Diffusion-based methods are longer than that of GAN-based methods. However in the tasks mentioned in this article, inference time is more important than training time. This is because training is a one-time offline process that can be completed in advance and has relatively flexible time, while reasoning directly affects clinical practical applications. So we only compared the inference time between different methods.
Comments 2: While the paper evaluates ScDiff's performance using quantitative metrics such as PSNR and SSIM, there is a lack of detailed discussion on its clinical significance. The authors should assess how doctors have applied ScDiff-generated images in real diagnoses or how it affects diagnostic accuracy. Comparing results with actual diagnoses or incorporating feedback from medical professionals would strengthen the model’s practical effectiveness.
Response 2: Thank for your suggestion. We have added a description of the clinical significance of the generated images in the last paragraph of Section 3.2.
Comments 3: The paper mentions a gradient explosion issue during the training of the Diffusion Model. Although the authors resolved this by adjusting the learning rate (lr = 5e-4) and applying dropout (0.2), there is no clear explanation for why these specific values were chosen. A more detailed justification of the parameter selection process is needed.
Response 3: Thank you for pointing this out. We have added a description of the parameter selection process in the second paragraph of Section 4.
Comments 4: The comparison between ScDiff and existing models (e.g., CycleGAN, DCLGAN, SynDiff) relies mainly on quantitative metrics such as PSNR and SSIM. To enhance visual evidence, the paper should highlight how specific artifacts (e.g., ring artifacts, linear artifacts) were corrected using zoomed-in images with annotated arrows.
Response 4: Based on your comments, we have added local zoomed-in images of the results obtained by different methods in Figure 5.
Comments 5: The SCConv (Spatial and Channel Reconstruction Convolution) module is a critical part of the model, but its implementation details are described rather briefly. The paper should provide a block diagram illustrating the SCConv module’s structure and operational principles in greater detail.
Response 5: Thanks for the comment. We have provided a detailed introduction in the SCConv module of section 2.2 and figure 1.

Round 2
Reviewer 1 Report
Comments and Suggestions for Authors
The paper can be accepted in this form
Comments on the Quality of English LanguageThe paper can be accepted in this form